# The influence of sex, gender, age, and ethnicity on psychosocial factors and substance use throughout phases of the COVID-19 pandemic

Lori A. Brotto[1,2]*, Kyle Chankasingh[3], Alexandra Baaske[2], Arianne Albert[2], Amy Booth[2], Angela Kaida[2,3], Laurie W. Smith[2], Sarai Racey[2], Anna Gottschlich[2], Melanie C. M. Murray[1,4], Manish Sadarangani[5], Gina S. Ogilvie[2,6], Liisa Galea[2,7]

1 Department of Obstetrics and Gynaecology, University of British Columbia, Vancouver, Canada,
2 Women's Health Research Institute, Vancouver, Canada, 3 Faculty of Health Sciences, Simon Fraser University, Burnaby, Canada, 4 Department of Medicine, University of British Columbia, Vancouver, Canada,
5 Department of Pediatrics, University of British Columbia, Vancouver, Canada, 6 School of Population and Public Health, University of British Columbia, Vancouver, Canada, 7 Department of Psychology, University of British Columbia, Vancouver, Canada

* lori.brotto@vch.ca

**Data Availability Statement:** Data cannot be shared publicly because of ethical restrictions. Data are available from the the UBC Research Ethics

## Abstract

### Objectives

The SARS-CoV-2 (COVID-19) pandemic has had profound physical and mental health effects on populations around the world. Limited empirical research has used a gender-based lens to evaluate the mental health impacts of the pandemic, overlooking the impact of public health measures on marginalized groups, such as women, and the gender diverse community. This study used a gender-based analysis to determine the prevalence of psychosocial symptoms and substance use (alcohol and cannabis use in particular) by age, ethnicity, income, rurality, education level, Indigenous status, and sexual orientation.

### Methods

Participants in the study were recruited from previously established cohorts as a part of the COVID-19 Rapid Evidence Study of a Provincial Population-Based Cohort for Gender and Sex (RESPPONSE) study. Those who agreed to participate were asked to self-report symptoms of depression, anxiety, pandemic stress, loneliness, alcohol use, and cannabis use across five phases of the pandemic as well as retrospectively before the pandemic.

### Results

For all psychosocial outcomes, there was a significant effect of time with all five phases of the pandemic being associated with more symptoms of depression, anxiety, stress, and loneliness relative to pre-COVID levels ($p < .0001$). Gender was significantly associated with all outcomes ($p < .0001$) with men exhibiting lower scores (i.e., fewer symptoms) than women and gender diverse participants, and women exhibiting lower scores than the

Board (contact via cwreb@bcchr.ubc.ca) for researchers who meet the criteria for access to confidential data.

**Funding:** Funding for this project was from a Michael Smith Foundation for Health Research Grant (19055) and a BC Women's Health Foundation Grant (LRZ30421) both awarded to Dr Lori Brotto and Dr. Gina S. Ogilvie.

**Competing interests:** The authors have declared that no competing interests exist.

gender diverse group. Other significant predictors were age (younger populations experiencing more symptoms, $p < .0001$), ethnicity (Chinese/Taiwanese individuals experiencing fewer symptoms, $p = .005$), and Indigenous status (Indigenous individuals experiencing more symptoms, $p < .0001$). Alcohol use and cannabis use increased relative to pre-pandemic levels, and women reported a greater increase in cannabis use than men ($p < .0001$).

## Conclusions

Our findings highlight the need for policy makers and leaders to prioritize women, gender-diverse individuals, and young people when tailoring public health measures for future pandemics.

## Introduction

In the first eighteen months of the SARS-CoV-2 (COVID-19) pandemic, there have been over 150 million cases and over 3 million deaths attributable to the upper respiratory virus [1]. More specifically, Canada has reached a stark milestone of one million cases and over 20,000 deaths in a little over a year (May 2021) [2]. Although the physical health effects of the virus tend to dominate the literature and the media, it is well established that outbreaks, including the current pandemic, have significant impacts on the mental health of those involved. For example, healthcare workers and patients affected by previous outbreaks such as SARS-CoV-1 [3], H1N1 influenza [4], and Ebola [5] have poorer psychosocial outcomes during the onset of societal alarm.

Public health measures put in place due to the COVID-19 pandemic have had a negative impact on the mental health of peoples worldwide [6,7]. Levels of depression [8], anxiety [8], loneliness [9], alcohol use [10,11], and cannabis use [12] have all increased relative to pre-pandemic levels. Additionally, there is mounting evidence highlighting the secondary effects of public health measures on specific populations during the pandemic [13]. For example, younger populations [14] and those of lower income [14] have experienced disproportionate psychosocial outcomes because of the COVID-19 pandemic.

There is a growing realization that a gender lens needs to be applied to COVID-19 research, not only regarding biomedical outcomes, but for psychosocial outcomes as well [15]. This aligns with increasing efforts, across North America, to include sex and gender based analyses in all research. Sex is defined as birth assignment and is usually established by genital anatomy at birth with female, male, and intersex as typical response options in queries about sex. Gender identity is defined as one's personal feelings about being a woman, man, transgender, gender-diverse individual, or another expression of gender that does not align with that person's birth assigned sex. When sex is considered in the context of psychosocial issues, it is well established that females are more likely to present symptoms of depression and anxiety in general [16], and face greater job losses than males during the COVID-19 pandemic [17]. Thus, it is not surprising that studies to date have found that females reported more anxiety, depressive symptoms, and post-traumatic stress symptoms relative to males during the COVID-19 pandemic [14,18–20]. Age also plays a large role in sex differences in the risk for neuropsychiatric disorders [21], but thus far the interaction between age and sex has received little attention with regards to how age may interact with sex to impact psychosocial outcomes throughout COVID-19.

In addition to the paucity of sex-based analyses, studies examining psychosocial outcomes from the standpoint of participants' self-identified gender are sparse. Most of the research on gender and the COVID-19 pandemic have compared responses between women and men, while ignoring the experiences of individuals who experience gender on a spectrum, beyond the binary classification of man and woman. A recent cross-sectional survey by Hawke et al. [22] found that despite no clear significant differences in mental health between cisgender, transgender and gender diverse youth before the pandemic, those identifying as gender diverse were two times more likely to report experiencing mental health challenges relative to the cisgender group during COVID-19. These findings were associated with an unmet need for mental health and substance use services. However, this study was limited to those aged 14–28, thereby reducing the generalizability of the findings to the larger population. To our knowledge, no studies have taken a gender-based approach to understanding the mental health sequelae of COVID-19 pandemic control measures across a general population sample. Given the current data as well as previous findings on the poor mental health outcomes of gender diverse individuals [23], focused empirical attention on this population is critical.

Other social determinants of health, including education, ethnicity, and income, impact physical and mental health outcomes [24] and have shaped the risk and consequences of COVID-19 in communities across North America [25]. Additionally, minority stress theory has posited that those who are part of more than one marginalized societal group may experience even greater health disparities [26]. Given this, we posited that it would be crucial to explore how gender interacted with these social determinants to influence mental health, in particular also because these social factors might moderate the effects of gender.

Many governments have tailored public health interventions throughout the pandemic based on infection incidence and hospitalization rates, resulting in a series of lockdowns (and prescribed regulations), followed by periods of relaxed restrictions, which have generated defined "phases" of the pandemic. While it is now widely known that lockdowns impact mental health [27], what remains unclear is how the tightening and easing of these social restrictions impacts psychosocial factors, by gender. As such, we sought to assess our psychosocial outcomes, cross-sectionally, across various phases of the pandemic retrospectively aligned with provincial changes in public health orders.

We were particularly interested in self-reported symptoms of depression, anxiety, stress, and loneliness, given recent reports of how these have worsened over the pandemic [8,9], and in addition to our primary interest in how gender impacted these outcomes, we also examined the interaction of gender with age, ethnicity, income, education, rurality, and sexual orientation. We predicted higher scores (i.e., more symptomatic) on depression, anxiety, stress, and loneliness in women and gender diverse individuals compared to men, and that this would be influenced by age, ethnicity, and income. We also hypothesized that during phases of increased social restriction psychosocial symptoms would increase. Our secondary interest was in self reports of alcohol and cannabis use, again given data on the pandemic's effects of substance use behaviors [10–12], and again, we explored the impact of gender, as well as how gender interacted with various social variables, on alcohol and cannabis use. Our analyses of gender and alcohol and cannabis use were exploratory. Using a large cohort of the general population in British Columbia (BC), we assessed participants cross-sectionally, and asked them to retrospectively report on these outcomes across different phases of the pandemic, which again corresponded to stages of pandemic control changes in the province of BC.

## Materials and methods

### Participant recruitment and study design

Participants, aged 25–69 years, were invited to participate in this study from previously established cohorts from research teams at the Women's Health Research Institute, representing both general and priority populations of BC who had consented to being contacted for future research [28]. The original cohorts represented healthy women aged 25–65, women living with HIV/AIDS, men and women over age 18 living with a complex chronic disease, and individuals over age 18 with pelvic pain and/or endometriosis [28]. The largest cohort was recruited to reflect a broad and representative sample of BC women. Participants were stratified into nine five-year age strata, and using a SARS-CoV-2 population seroprevalence of 2% (±1, 95% CI), the target recruitment for each stratum was 750. The seroprevalence statistic was used to target recruitment for analyses in a separate manuscript.

Those identified as potentially eligible from the established cohorts (*Index Participants)* were sent an email invitation to participate via an online survey. To improve the sample size and gender diversity of the study sample, Index Participants were asked to pass the invitation on to one household member who identified as a different gender as the respondent *(Household Participants)*. All potential participants were sent up to three email reminders to participate in the study. The inclusion criteria were: current residents of BC, aged 25–69, any gender, and able and willing to fill out the online survey in English. Ethics approval was obtained from the BC Children's and Women's Research Ethics board, and all participants provided consent to participate. Survey responses were collected anonymously, with the exception of three-digit postal codes, which were used to determine rurality for analyses.

After two months of data collection from existing research cohorts, recruitment was expanded in order to meet our target sample of n = 750 per age cohort. This expanded recruitment included participants obtained from public recruitment through the REACH BC platform, social media (i.e., Facebook, Twitter and Instagram), posts on the Women's Health Research Institute website (www.whri.org) and engagement of community groups who are affiliated with the Women's Health Research Institute, a provincial research institute focused on gender and women's health. All respondents in the study were invited to enter a draw to win a $100 e-gift card for completing the survey. Recruitment was continued until a target of n = 750 was reached for each of the nine age-based strata, with the exception of the 25–29 year age group. Recruitment was open from August 20, 2020 –March 1, 2021.

### Survey design and measures

The survey was tested for face validity, pilot tested, and a final version was designed using REDCap (Research Electronic Data Capture) [29]. While the survey consisted of multiple modules, this study focuses solely on the outcomes from the psychosocial module, which included questions about mental health outcomes such as depression, anxiety, stress, loneliness, alcohol use, and cannabis use.

Demographic information was collected from all respondents including age, sex, gender, sexual orientation, ethnicity, Indigenous status, income, education level, if the participant was currently a student and rurality by postal code. Sex referred specifically to the sex assigned at birth and included the option of male, female or intersex. Gender referred to the respondent's current gender identity and included the options man, woman, or another option grouping non-binary, transgender, GenderQueer, agender or any other similar identity together. Sexual orientation options included asexual, bisexual, demisexual, gay/lesbian, heterosexual, or pansexual. Participants were given the option to identify as the following ethnicities: White,

Chinese/Taiwanese, Black (African, Caribbean, or Other), South Asian (e.g., Indian, Bangla-deshi, Pakistani, Punjabi, and Sri Lankan), and several other ethnicities who were analyzed an "Other" category. Indigenous status was assessed separately from ethnicity. Self-reporting of Indigenous status provided participants the option to identify as First Nation, Metis, Inuit, non-status First Nations, other Indigenous or not Indigenous, and they were then asked about Two Spirit identity. Rurality was determined based on three-digit postal codes and were classi-fied into one of the follow categories: census metropolitan area, strong metropolitan influence zone, moderate metropolitan zone, or weak to no metropolitan influence zone.

The study design was cross-sectional in nature, whereby participants completed the survey at one time point. However, several questions asked participants to retrospectively refer to dif-ferent periods of time: pre-pandemic (before March 2020) as well as across five different phases of the pandemic in BC that corresponded with changes in the public health orders regarding social distancing in the province of BC. In the original version of the survey, Phase 1 lasted from mid-March 2020 to mid-May 2020, Phase 2 lasted from mid-May 2020 to mid-June 2020, and Phase 3 lasted from mid-June 2020 until the end of November 2020. Given that data collection continued past November 2020, we added Phases 4 and 5, as well as modified dates for Phases 2 and 3 (mid-May to end of August 2020; denoted by Phase 2/3_2). Phase 4 lasted from September 2020 to the end of October 2020 and Phase 5 lasted from November 2020 to the date our survey closed (March 1, 2021). We have included a S1 Table that explains the public health recommendations in more detail, through every phase of the pandemic in BC.

**Depression.**    Depression was measured across the phases of the pandemic using the Patient Health Questionnaire (PHQ-9). The PHQ-9 questionnaire was used to measure self-reported symptoms of depression on a Likert scale from zero (not at all) to three (nearly every-day). Scores for this questionnaire range from 0–27 with a score of 0–4 indicating minimal depression, 5–14 indicating mild to moderate depression and 15–27 indicating moderately severe to severe depression [30]. The PHQ-9 has been validated across age and gender, as well as among diverse populations [30,31]. Internal consistency across data collection and Cron-bach's alpha for the PHQ-9 in the current sample was very good at $\alpha = 0.848$.

**Anxiety.**    Anxiety was measured across the phases of the pandemic using the Generalized Anxiety Disorder questionnaire (GAD-7). The GAD-7 was provided to respondents to self-report feelings of anxiety on a Likert scale from zero (not at all) to three (nearly everyday). Scores for this questionnaire range from 0–21 with scores above 10 indicating a clinical diag-nosis for anxiety [32]. The GAD-7 has been validated in the general population and is fre-quently used in primary care settings to screen for anxiety symptoms[33]. Internal consistency across data collection and Cronbach's alpha for the GAD-7 questionnaire in the current sam-ple was very good at $\alpha = 0.889$.

**Pandemic stress.**    General pandemic stress was measured across the phases of the pan-demic using the CoRonavIruS Health Impact Survey (CRISIS) V0.3. This survey was devel-oped and validated early in the COVID-19 pandemic to provide a general measure of mental distress and resilience [34]. The CRISIS is found to have strong validity and reliability, and has been recommended for use in population-based studies of mental health during COVID-19. Participants were asked to self-report feelings of stress on a Likert scale from one (not at all) to five (extremely). Scores for this questionnaire range from 10–50 with higher scores indicating greater COVID-related stress. Internal consistency across data collection and Cronbach's alpha for CRISIS in the current sample was very good at $\alpha = 0.882$.

**Loneliness.**    Loneliness was also measured across the phases of the pandemic where respondents were asked to self-report feelings of loneliness on a Likert scale from one (not lonely at all) to five (extremely lonely). This item was taken from the validated Coronavirus

Health and Impact Survey (CRISIS), where individual items on the CRISIS have been found to have high Intraclass Correlation Coefficients [34]. Previous studies have found loneliness to be positively correlated with both PHQ-9 and GAD-7 scores [35] and to be a significant predictor of suicide [36,37].

**Alcohol use.** Change in alcohol use was asked for all post-COVID time points (i.e., Has your consumption of alcohol changed since March 2020?). Change in alcohol use was defined as "none" (which included no alcohol use, decreased alcohol use, and same alcohol use) vs. increased alcohol use. Therefore, a single, non-time-varying alcohol change variable was created and used to compare the retrospective responses across the different time points, with time.

**Cannabis use.** Change in cannabis use was asked for all post-COVID time points (Has your consumption of cannabis changed since March 2020?). As with alcohol, change in cannabis use was defined as "none" (which included no cannabis use, decreased cannabis use, and same cannabis use) vs. increased cannabis use. A single, non-time-varying cannabis change variable was created and used in a longitudinal model with time.

## Statistical analyses

Analyses were carried out using R v.4.0.3. Analyses of psychosocial outcomes across the pandemic control phases were conducted using mixed-effects linear regression models with individual and household IDs as random effects. This allows for correlations among individuals in the same household, and separately, correlations over time among responses within the same individual, allowing for a longitudinal assessment. We included pairwise interactions to assess non-additive effects between age and gender, and age and ethnicity, sexual orientation, income, and Indigenous status. Significance was assessed using likelihood-ratio tests, and interactions were removed from the models if non-significant at $p < .05$. Post-hoc pairwise tests were conducted to further explore main or interaction effects with Bonferroni correction for multiple tests.

To explore associations between increase in alcohol and cannabis use with sex/gender and other demographic variables we used mixed-effects logistic regressions with household ID as a random effect. We also examined increase in alcohol and cannabis use and psychosocial outcomes across the phases as described above. Interactions and post-hoc tests were handled as above. Missing data were excluded from analyses.

## Results

### Survey participants

Between August 2020 and March 2021, 16,056 survey invites were emailed to prospective Index Participants and 1,872 participants were recruited from the public, for a total of 17,928 prospective participants. Of these participants, a total of 5,415 responded to the invitation to participate in the study and met the analysis inclusion criteria (Fig 1). Of these participants, 1,434 forwarded the survey invitation to a household member of a different sex or gender and we received 661 participants via this method. The present analyses includes the 6,076 Index and Household participants who completed psychosocial measures of anxiety, depression, stress, and loneliness.

### Demographic characteristics of participants

A total of 6,426 individuals responded to the question about sex; there were n = 820 males (12.7%) and n = 5,606 females (87.1%). A total of 6,076 responded to the question about

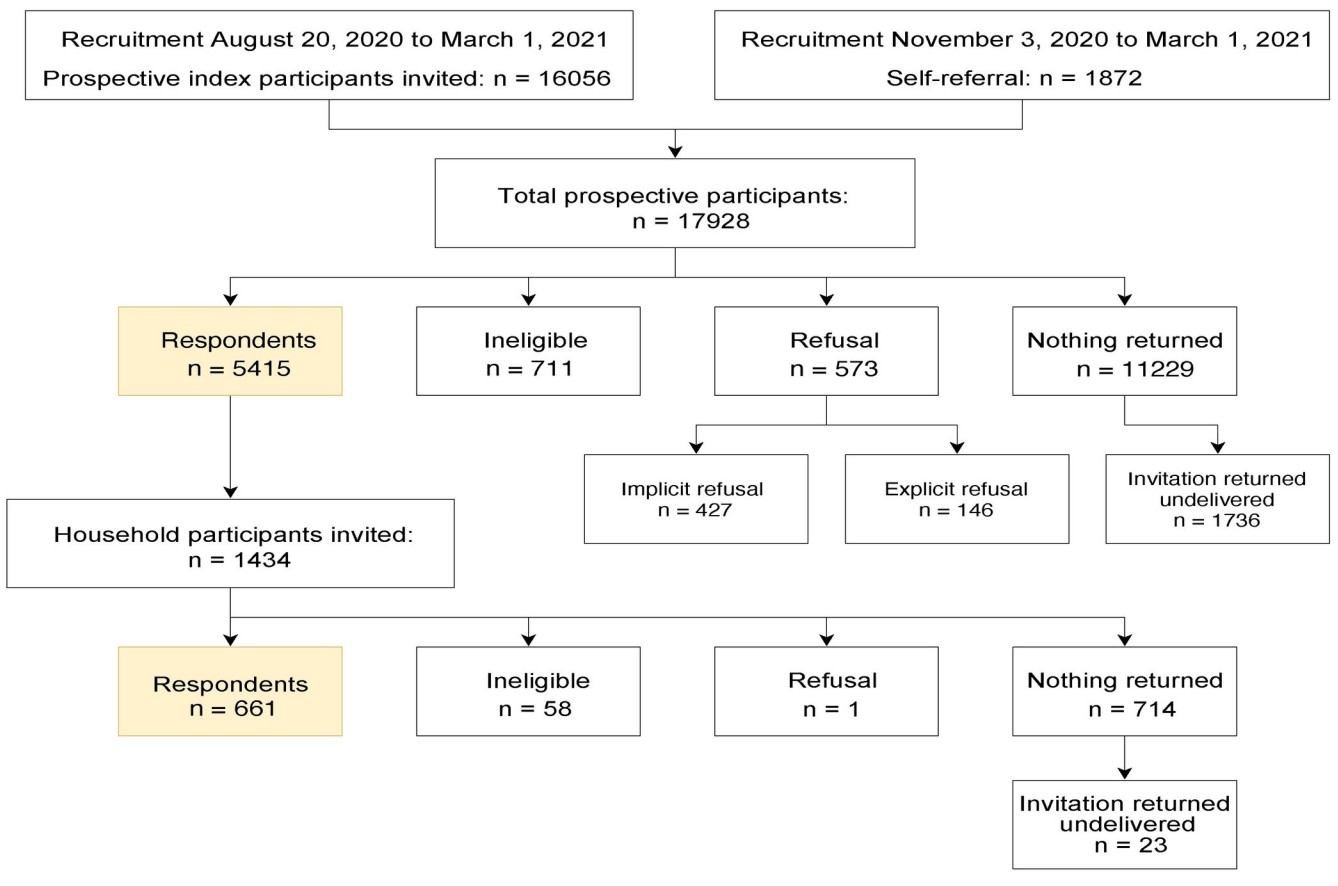

**Fig 1. A flow diagram of prospective participants and respondents to the study.**

gender; including men (n = 750; 12.3%), women (n = 5,254; 86.4%), and gender diverse (n = 72; 1.2%) individuals. Table 1 presents the demographic characteristics of the sample by gender, according to women, men, and gender diverse.

### Effect of pandemic phase, age, ethnicity and gender and sex on psychosocial outcomes

Controlling for household income, we found no significant interactions between age and gender, age and sex, age and ethnicity or rurality on any of the psychosocial measures. For all psychosocial outcomes, there was a significant relationship with pandemic phase (all $p < .0001$, Table 2), with the greatest increases in mental health symptoms in Phase 1 compared to pre-COVID. The scores in all subsequent phases remained significantly higher (i.e., more symptoms) than in the pre-COVID phase across all outcomes (Figs 2–5, Table 2). Gender was significantly associated with all outcomes (all $p < .0001$, Figs 2–5, Table 2), and pairwise comparisons showed that men had lower scores than both women and gender-diverse participants, while women had lower scores than the gender-diverse participants. Age was significantly negatively associated with all the outcomes, with older participants having lower scores on average (i.e., fewer psychosocial symptoms) ($p < .0001$, Table 2). Finally, there was a significant relationship between ethnicity and all outcomes (GAD-7 and PHQ-9 $p < .0001$, CRISIS and Loneliness $p = .005$, Table 2), with scores lower in Chinese/Taiwanese participants compared to the White, South Asian, and Other ethnicity participants. When sex was included in

**Table 1. Demographic information of survey respondents.**

| Total N = 6,076 | Women n = 5,254 | Men n = 750 | Gender Diverse n = 72 |
|---|---|---|---|
| **Sex N(%)** | | | |
| **Male** | 9 (0.2%) | 744 (99.2%) | 6 (8.3%) |
| **Female** | 5,243 (99.8%) | 5 (0.7%) | 62 (86.1%) |
| **Age *M (SD)*** | 48.5 (±12.0) | 48.5 (±12.2) | 42.4 (±11.6) |
| 25–29 | 348 (6.6%) | 68 (9.1%) | 8 (11.1%) |
| 30–39 | 1,075 (20.5%) | 133 (17.7%) | 29 (40.3%) |
| 40–49 | 1,291 (24.6%) | 164 (21.9%) | 14 (19.4%) |
| 50–59 | 1,333 (25.4%) | 212 (28.3%) | 11 (15.3%) |
| 60–69 | 1,207 (23.0%) | 173 (23.1%) | 10 (13.9%) |
| **Sexual Orientation (%)** | | | |
| Heterosexual | 4,480 (85.3%) | 674 (89.9%) | 7 (9.7%) |
| Non-Heterosexual | 757 (14.4%) | 74 (9.9%) | 65 (90.3%) |
| **Ethnicity (%)** | | | |
| White | 4,265 (81.2%) | 609 (81.2%) | 44 (61.1%) |
| Black | 28 (0.5%) | 6 (0.8%) | 1 (1.4%) |
| Chinese/Taiwanese | 311 (5.9%) | 40 (5.3%) | 2 (2.8%) |
| South Asian | 123 (2.3%) | 9 (1.2%) | 1 (1.4%) |
| Other ethnicity | 504 (9.6%) | 81 (10.8%) | 20 (27.8%) |
| **Indigenous Status (%)** | | | |
| Indigenous | 166 (3.2%) | 27 (3.6%) | 11 (15.3%) |
| Not Indigenous | 4,830 (91.9%) | 696 (92.8%) | 59 (81.9%) |
| **Current Student (%)** | | | |
| Yes | 311 (5.9%) | 37 (4.9%) | 14 (19.4%) |
| No | 4,939 (94.0%) | 711 (94.8%) | 58 (80.6%) |
| **Level of education (%)** | | | |
| High school or less | 641 (12.2%) | 92 (12.3%) | 13 (18.1%) |
| More than high school | 4,605 (87.6%) | 658 (87.7%) | 58 (80.6%) |
| **Current Household Income (%)** | | | |
| <$10,000 –$20,000 | 128 (2.4%) | 14 (1.9%) | 7 (9.7%) |
| $20,000 –$40,000 | 269 (5.1%) | 23 (3.1%) | 14 (19.4%) |
| $40,000 –$60,000 | 458 (8.7%) | 32 (4.3%) | 10 (13.9%) |
| $60,000 –$80,000 | 488 (9.3%) | 56 (7.5%) | 5 (6.9%) |
| $80,000 –$100,000 | 642 (12.2%) | 67 (8.9%) | 8 (11.1%) |
| $100,000 –$150,000 | 1,104 (21.0%) | 156 (20.8%) | 14 (19.4%) |
| >$150,000 | 1,273 (24.2%) | 304 (40.5%) | 6 (8.3%) |
| **Rurality** | | | |
| Census metropolitan area | 4,957 (94.3%) | 716 (95.5%) | 70 (97.3%) |
| Strong metropolitan influence zone | 79 (1.4%) | 7 (0.9%) | 0 (0.0%) |
| Moderate metropolitan influence zone | 116 (2.0%) | 8 (1.1%) | 1 (1.4%) |
| Weak to No metropolitan influence zone | 46 (0.8%) | 5 (0.67%) | 0 (0.0%) |

*Note*: Values do not add up to 100% due to missing data.

*M* refers to the mean age and *(SD)* standard deviation of participants.

the model in place of gender, there were no differences to the findings, indicating the overlap in our participants self-reported sex and gender. Given our intention to explore outcomes separately for gender-diverse individuals, all subsequent analyses were done by gender (not sex).

**Table 2. Impact of sociodemographic factors and pandemic phase on psychosocial outcomes.**

| | Anxiety (GAD-7) | | Depression (PHQ-9) | | Pandemic Stress (CRISIS) | | Loneliness | |
|---|---|---|---|---|---|---|---|---|
| **Predictors** | *Estimates* | *95% CI* | *Estimates* | *95% CI* | *Estimates* | *95% CI* | *Estimates* | *95% CI* |
| (Intercept) | 8.72** | 7.97 – 9.47 | 9.58** | 8.81 – 10.35 | 29.35** | 28.18 – 30.51 | 2.56** | 2.40 – 2.73 |
| **Age (per year increase)** | -0.1** | -0.10 – -0.09 | -0.08** | -0.09 – -0.07 | -0.15** | -0.16 – -0.13 | -0.01** | -0.01 – -0.01 |
| **Phase** | | | | | | | | |
| Pre-COVID | *Reference** | | *Reference** | | *Reference** | | *Reference** | |
| Phase 1 | 2.92 | 2.82 – 3.01 | 2.38 | 2.28 – 2.48 | 7.31 | 7.13 – 7.49 | 0.62 | 0.59 – 0.64 |
| Phase 2/3 | 2.09 | 1.98 – 2.19 | 1.98 | 1.88 – 2.09 | 5.34 | 5.15 – 5.53 | 0.47 | 0.45 – 0.50 |
| Phase 2/3_2 | 1.61 | 1.40 – 1.83 | 1.56 | 1.34 – 1.78 | 4.42 | 4.03 – 4.81 | 0.42 | 0.37 – 0.48 |
| Phase 4 | 2.16 | 1.95 – 2.37 | 2.44 | 2.23 – 2.66 | 6.08 | 5.69 – 6.47 | 0.63 | 0.58 – 0.69 |
| Phase 5 | 2.53 | 2.32 – 2.74 | 2.99 | 2.78 – 3.21 | 7.31 | 6.92 – 7.71 | 0.91 | 0.85 – 0.96 |
| **Household Income** | | | | | | | | |
| <$10K to $20K | *Reference** | | *Reference** | | *Reference** | | *Reference** | |
| $20K to $40K | 0.51 | -0.23 – 1.25 | -0.56 | -1.33 – 0.20 | -0.29 | -1.45 – 0.86 | -0.2 | -0.37 – -0.04 |
| $40K to $60K | 0.04 | -0.66 – 0.73 | -0.93 | -1.65 – -0.21 | -1.12 | -2.20 – -0.04 | -0.31 | -0.46 – -0.15 |
| $60K to $80K | -0.62 | -1.31 – 0.07 | -1.93 | -2.65 – -1.22 | -2.56 | -3.63 – -1.49 | -0.47 | -0.62 – -0.32 |
| $80K to $100K | -1.03 | -1.70 – -0.35 | -2.43 | -3.13 – -1.74 | -2.99 | -4.03 – -1.95 | -0.57 | -0.71 – -0.42 |
| $100K to $150K | -1.13 | -1.78 – -0.49 | -2.66 | -3.33 – -1.99 | -3.47 | -4.47 – -2.47 | -0.69 | -0.83 – -0.55 |
| >$150K | -1.55 | -2.19 – -0.91 | -3.38 | -4.04 – -2.71 | -4.42 | -5.41 – -3.42 | -0.8 | -0.94 – -0.66 |
| **Gender** | | | | | | | | |
| Women | *Reference** | | *Reference** | | *Reference** | | *Reference** | |
| Men | -1.44 | -1.75 – -1.13 | -1.25 | -1.57 – -0.93 | -2.28 | -2.75 – -1.80 | -0.22 | -0.29 – -0.15 |
| Gender Diverse | 1.67 | 0.71 – 2.64 | 2.09 | 1.11 – 3.07 | 3.21 | 1.72 – 4.71 | 0.34 | 0.12 – 0.55 |
| **Ethnicity** | | | | | | | | |
| White | *Reference** | | *Reference** | | *Reference** | | *Reference** | |
| Black | -0.4 | -1.74 – 0.95 | 0.51 | -0.87 – 1.89 | -0.19 | -2.32 – 1.94 | 0.06 | -0.24 – 0.36 |
| Chinese/Taiwanese | -0.98 | -1.45 – -0.52 | -1.36 | -1.84 – -0.88 | -1.14 | -1.87 – -0.42 | -0.18 | -0.28 – -0.08 |
| South Asian | 0.43 | -0.33 – 1.18 | 0.08 | -0.69 – 0.85 | 1.01 | -0.18 – 2.19 | 0.09 | -0.07 – 0.26 |
| Other | 0.38 | 0.02 – 0.73 | 0.14 | -0.22 – 0.51 | 0.34 | -0.21 – 0.89 | 0.03 | -0.05 – 0.11 |
| **Indigenous Status Across Phases** | | | | | | | | |
| Pre-COVID | *Reference** | | *Reference** | | *Reference** | | *Reference** | |
| Phase 1 | 2.87 | 2.77–2.97 | 2.33 | 2.23–2.44 | 7.26 | 7.08–7.45 | 0.60 | 0.58–0.63 |
| Phase 2/3 | 2.06 | 1.95–2.17 | 1.94 | 1.83–2.05 | 5.28 | 5.08–5.47 | 0.46 | 0.43–0.49 |
| Phase 2/3_2 | 1.57 | 1.34–1.79 | 1.46 | 1.23–1.69 | 4.31 | 3.90–4.72 | 0.41 | 0.35–0.46 |
| Phase 4 | 2.14 | 1.91–2.36 | 2.36 | 2.13–2.59 | 5.98 | 5.57–6.39 | 0.61 | 0.55–0.67 |
| Phase 5 | 2.53 | 2.30–2.75 | 2.94 | 2.71–3.17 | 7.34 | 6.93–7.75 | 0.88 | 0.82–0.94 |
| **Non-Heterosexual Orientation Across Phases** | | | | | | | | |
| Pre-COVID | *Reference** | | *Reference** | | *Reference** | | *Reference** | |
| Phase 1 | 2.81 | 2.70–2.92 | 2.28 | 2.17–2.38 | 7.21 | 7.01–7.40 | 0.58 | 0.55–0.61 |
| Phase 2/3 | 2.00 | 1.89–2.12 | 1.89 | 1.77–2.00 | 5.25 | 5.04–5.46 | 0.45 | 0.42–0.48 |
| Phase 2/3_2 | 1.56 | 1.33–1.80 | 1.45 | 1.21–1.69 | 4.36 | 3.93–4.79 | 0.42 | 0.36–0.48 |
| Phase 4 | 2.17 | 1.93–2.40 | 2.40 | 2.16–2.64 | 6.13 | 5.70–6.56 | 0.63 | 0.57–0.70 |
| Phase 5 | 2.58 | 2.35–2.81 | 2.96 | 2.72–3.20 | 7.41 | 6.97–7.84 | 0.91 | 0.85–0.97 |
| **Marginal R² /Conditional R²** | 0.173 / 0.716 | | 0.164 / 0.720 | | 0.255 / 0.674 | | 0.149 / 0.623 | |

*CI* refers to confidence intervals for the adjusted estimates. Pre-COVID: Prior to mid-March 2020; Phase 1: Mid-March 2020 to mid-May 2020; Phase: 2/3: Mid-May 2020 to November 2020; Phase 2/3_2: Mid-May 2020 to August 2020; Phase 4: September 2020 to October 2020; Phase 5: November 2020 to March 1, 2021. GAD-7: Generalized Anxiety Disorder measure; PHQ-9: Patient Health Questionnaire; CRISIS: CoRonavIruS Health Impact Survey; Loneliness was measured using a single item.

*$p = .005$

**$p < .0001$.

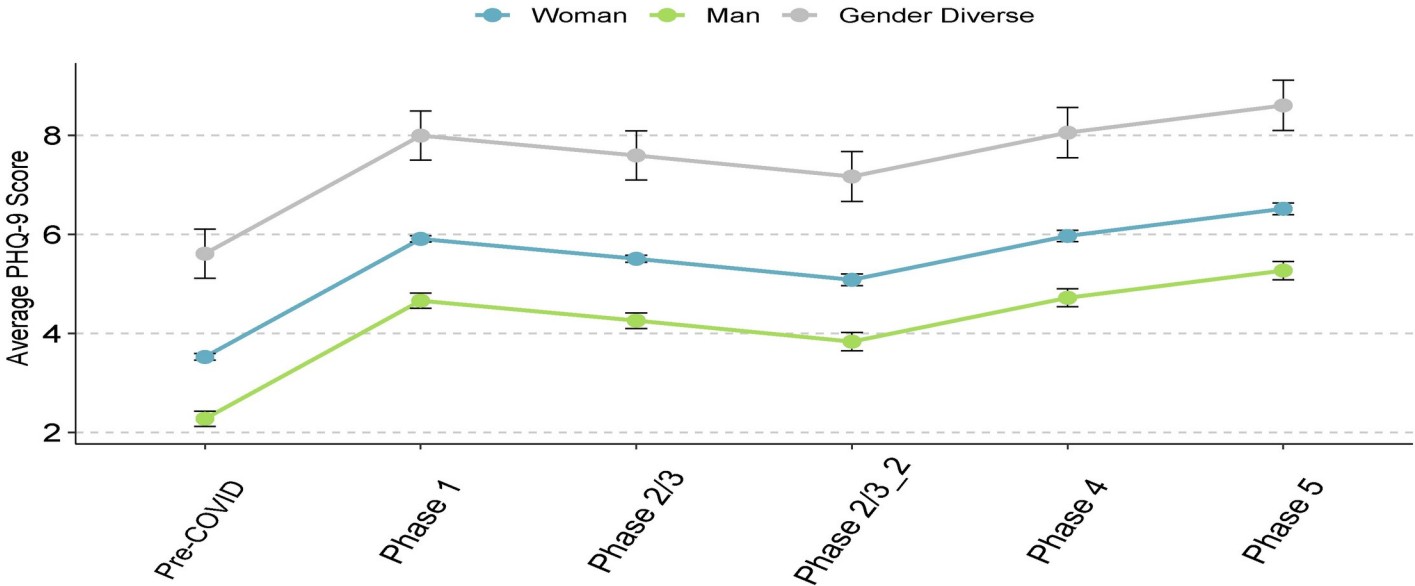

**Fig 2. Depressive symptoms, as measured by the PHQ-9, across phases of the pandemic.** Data points refer to mean scores of the given psychosocial measure, error bars refer to the standard error. Pre-COVID: Prior to mid-March 2020; Phase 1: Mid-March 2020 to mid-May 2020; Phase: 2/3: Mid-May 2020 to November 2020; Phase 2/3_2: Mid-May 2020 to August 2020; Phase 4: September 2020 to October 2020; Phase 5: November 2020 to March 1, 2021.

## Psychosocial outcomes by indigenous status

Controlling for household income, there was no significant interaction between Indigenous status and age or gender. There was a significant interaction between Indigenous status and time for all four psychosocial outcomes ($p < .0001$, Table 2) and follow-up post-hoc pairwise tests suggest that at all time points except pre-COVID, those who identified as Indigenous had

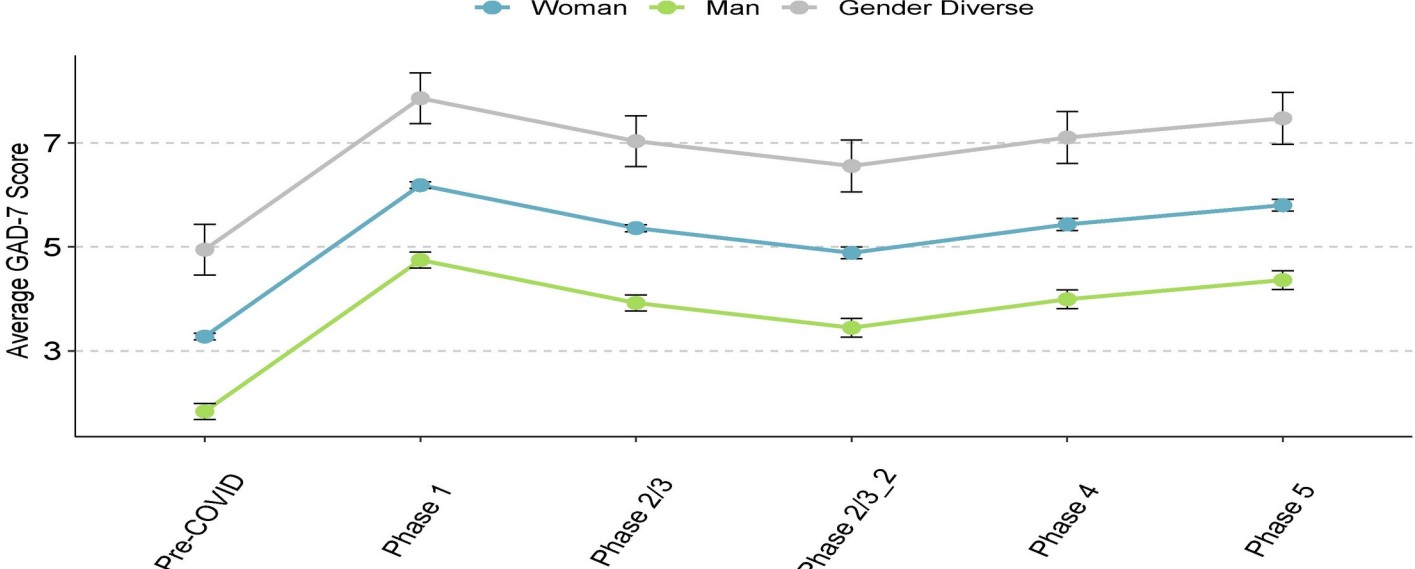

**Fig 3. Anxiety symptoms, as measured by the GAD-7, across phases of the pandemic.** Data points refer to mean scores of the given psychosocial measure, error bars refer to the standard error. Pre-COVID: Prior to mid-March 2020; Phase 1: Mid-March 2020 to mid-May 2020; Phase: 2/3: Mid-May 2020 to November 2020; Phase 2/3_2: Mid-May 2020 to August 2020; Phase 4: September 2020 to October 2020; Phase 5: November 2020 to March 1, 2021.

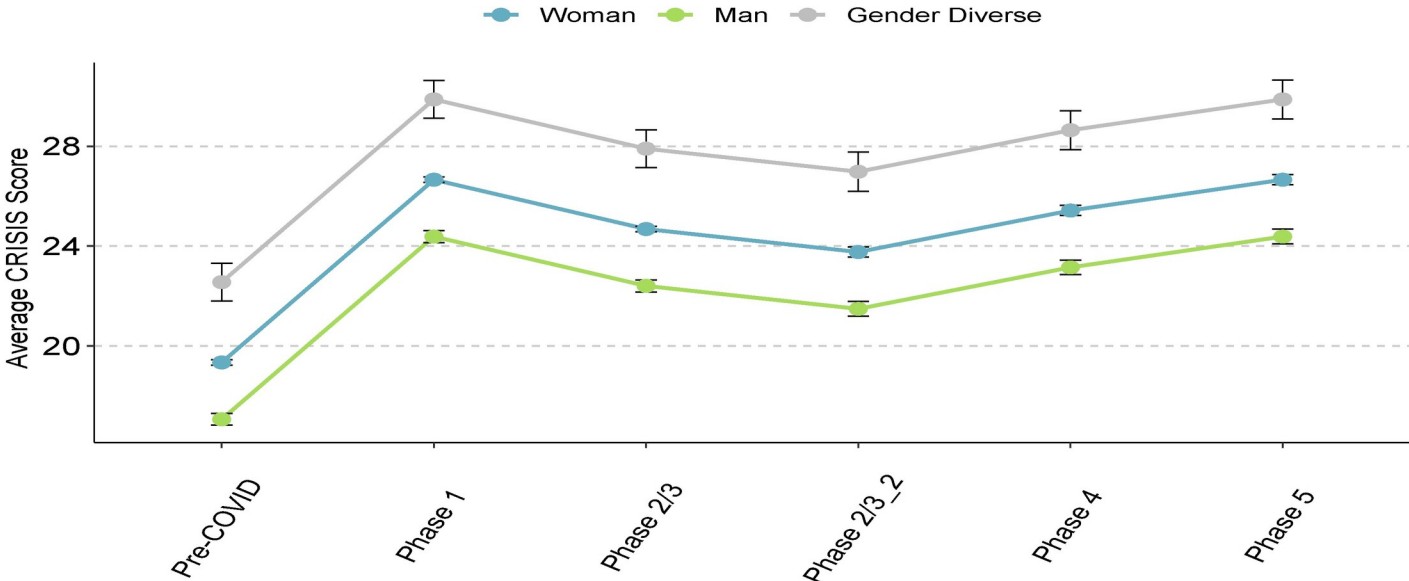

**Fig 4. Pandemic stress symptoms, as measured by the CoRonavIruS Health Impact Survey (CRISIS), across phases of the pandemic.** Data points refer to mean scores of the given psychosocial measure, error bars refer to the standard error. Pre-COVID: Prior to mid-March 2020; Phase 1: Mid-March 2020 to mid-May 2020; Phase: 2/3: Mid-May 2020 to November 2020; Phase 2/3_2: Mid-May 2020 to August 2020; Phase 4: September 2020 to October 2020; Phase 5: November 2020 to March 1, 2021.

significantly higher GAD-7, PHQ-9, CRISIS, and loneliness scores (i.e., more mental health symptoms) than those who did not identify as Indigenous.

### Psychosocial outcomes by sexual orientation

Across all outcomes, the non-heterosexual group (which included asexual, bisexual, demisexual, gay/lesbian, pansexual, and other) had significantly more mental health symptoms than

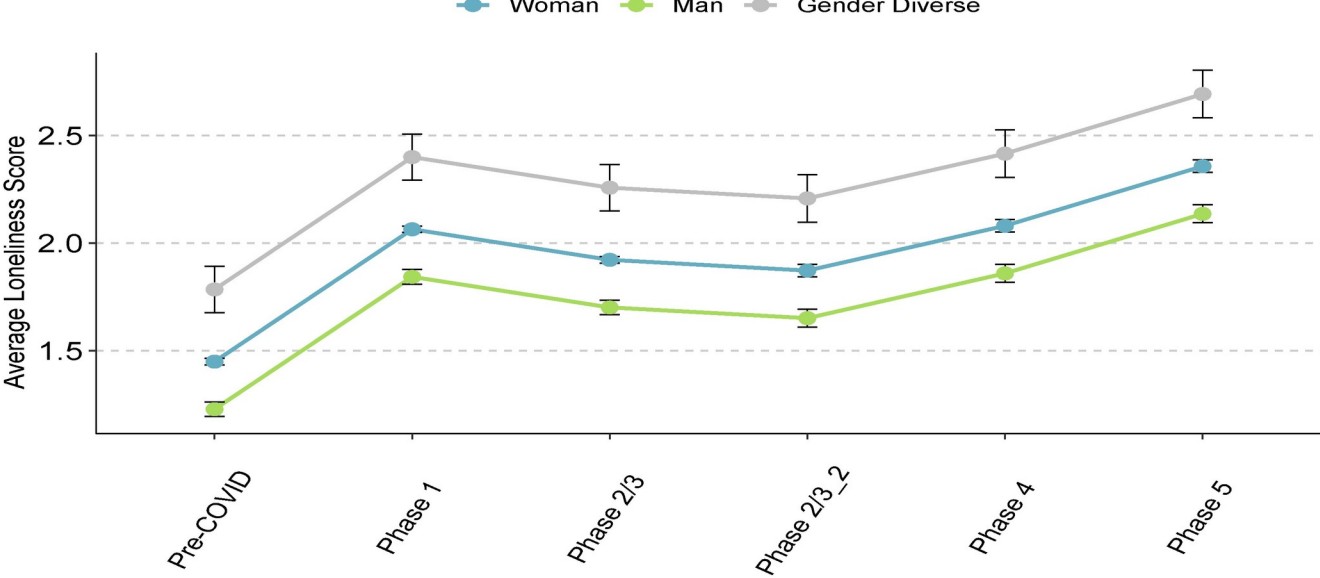

**Fig 5. Loneliness symptoms across phases of the pandemic.** Data points refer to mean scores of the given psychosocial measure, error bars refer to the standard error. Pre-COVID: Prior to mid-March 2020; Phase 1: Mid-March 2020 to mid-May 2020; Phase: 2/3: Mid-May 2020 to November 2020; Phase 2/3_2: Mid-May 2020 to August 2020; Phase 4: September 2020 to October 2020; Phase 5: November 2020 to March 1, 2021.

**Table 3. Changes in alcohol and cannabis use across sociodemographic factors.**

| Predictors | Change in Alcohol Use | | Change in Cannabis Use | |
|---|---|---|---|---|
| | *OR* | *95% CI* | *OR* | *95% CI* |
| **Age** | 0.98** | 0.97 – 0.98 | 0.97** | 0.96 – 0.99 |
| **Household income** | | | | |
| <$10K to $20K | *Reference** | | *Reference* | |
| $20K to $40K | 1.16 | 0.64 – 2.11 | 1.5 | 0.67 – 3.37 |
| $40K to $60K | 1.19 | 0.68 – 2.08 | 1.32 | 0.60 – 2.91 |
| $60K to $80K | 1.13 | 0.66 – 1.95 | 1.67 | 0.76 – 3.70 |
| $80K to $100K | 1.28 | 0.75 – 2.18 | 1.39 | 0.64 – 3.03 |
| $100K to $150K | 1.45 | 0.86 – 2.43 | 1.24 | 0.59 – 2.61 |
| >$150K | 1.62 | 0.97 – 2.70 | 1.17 | 0.55 – 2.48 |
| **Rurality** | | | | |
| Census metropolitan area | *Reference** | | *Reference* | |
| Strong metropolitan influence zone | 0.93 | 0.48 – 1.82 | 0.61 | 0.19 – 2.00 |
| Moderate metropolitan influence zone | 1.42 | 0.85 – 2.36 | 0.84 | 0.37 – 1.91 |
| Weak to No metropolitan influence zone | 0.24 | 0.07 – 0.81 | 2.15 | 0.48 – 9.61 |
| **Gender** | | | | |
| Women | *Reference* | | *Reference** | |
| Men | 0.86 | 0.69 – 1.06 | 0.66 | 0.43 – 1.02 |
| Gender Diverse | 0.72 | 0.32 – 1.61 | 0.84 | 0.34 – 2.11 |
| **Ethnicity** | | | | |
| White | *Reference* | | *Reference* | |
| Black | 1.17 | 0.49 – 2.82 | 2.42 | 0.66 – 8.89 |
| Chinese/Taiwanese | 0.68 | 0.45 – 1.02 | 0.57 | 0.24 – 1.33 |
| South Asian | 0.89 | 0.52 – 1.53 | 0.38 | 0.10 – 1.40 |
| Other | 0.94 | 0.72 – 1.22 | 1.56 | 1.02 – 2.37 |

*OR* refers to the odds ratio, *CI* refers to the confidence intervals for the *OR*.

* *p* = .03

** *p* < .001.

the heterosexual group for all phases, and the magnitude of the difference between the groups was largest in Phase 1 of the pandemic.

## Associations between psychosocial outcomes and alcohol by gender

A total of 23.3% of the sample reported an increase in alcohol use. Increased alcohol use was negatively associated with age (*p* < .001, Table 3), with older participants having lower odds of increased alcohol use. There was no significant difference among genders in the odds of increased alcohol use, but there was a trend of increasing odds as household income increased. Additionally, those residing in census metropolitan areas were found to have increased their alcohol use relative to those outside of these dense urban areas (*p* = .03, Table 3).

Controlling for household income, and across all psychosocial outcomes, there was no interaction between gender and increased alcohol use, suggesting that the differences among genders in these psychosocial variables was the same between those who did and did not increase alcohol use since the start of the pandemic (Table 4). There was a significant interaction between increased alcohol use and pandemic phase (all *p* < .0001, Table 4). Pairwise tests indicated that at all phases, with the exception of pre-COVID, those who reported increased alcohol use had significantly more psychosocial symptoms on all measures (*p* < .0001, Table 4).

**Table 4. Psychosocial outcomes and sociodemographic factors for those that reported an increase in alcohol use.**

| Predictors | Anxiety (GAD-7) Estimates | 95% CI | Depression (PHQ-9) Estimates | 95% CI | Pandemic Stress (CRISIS) Estimates | 95% CI | Loneliness Estimates | 95% CI |
|---|---|---|---|---|---|---|---|---|
| (Intercept) | 8.64* | 7.91 – 9.38 | 9.39* | 8.63 – 10.15 | 29.39* | 28.25 – 30.54 | 2.56* | 2.40 – 2.72 |
| **Age (per year increase)** | -0.09* | -0.10 – -0.08 | -0.07* | -0.08 – -0.06 | -0.14* | -0.16 – -0.13 | -0.01* | -0.01 – -0.01 |
| **Phase** | | | | | | | | |
| Pre-COVID | *Reference** | | *Reference** | | *Reference** | | *Reference** | |
| Phase 1 | 2.64 | 2.53 – 2.75 | 2.13 | 2.01 – 2.24 | 6.74 | 6.53 – 6.94 | 0.58 | 0.55 – 0.60 |
| Phase 2/3 | 1.9 | 1.78 – 2.02 | 1.77 | 1.65 – 1.89 | 4.91 | 4.69 – 5.13 | 0.44 | 0.41 – 0.47 |
| Phase 2/3_2 | 1.42 | 1.17 – 1.66 | 1.36 | 1.11 – 1.61 | 4.01 | 3.56 – 4.47 | 0.38 | 0.32 – 0.45 |
| Phase 4 | 1.91 | 1.67 – 2.16 | 2.26 | 2.01 – 2.51 | 5.62 | 5.16 – 6.07 | 0.59 | 0.52 – 0.65 |
| Phase 5 | 2.22 | 1.98 – 2.47 | 2.71 | 2.46 – 2.96 | 6.78 | 6.33 – 7.23 | 0.89 | 0.82 – 0.95 |
| **Household Income** | | | | | | | | |
| <$10K to $20K | *Reference** | | *Reference** | | *Reference** | | *Reference** | |
| $20K to $40K | 0.51 | -0.23 – 1.25 | -0.58 | -1.34 – 0.18 | -0.31 | -1.46 – 0.84 | -0.21 | -0.37 – -0.05 |
| $40K to $60K | -0.02 | -0.71 – 0.67 | -0.97 | -1.69 – -0.25 | -1.17 | -2.25 – -0.09 | -0.31 | -0.46 – -0.16 |
| $60K to $80K | -0.69 | -1.37 – 0.00 | -1.97 | -2.68 – -1.26 | -2.63 | -3.70 – -1.56 | -0.48 | -0.63 – -0.33 |
| $80K to $100K | -1.07 | -1.74 – -0.40 | -2.46 | -3.15 – -1.77 | -3.02 | -4.06 – -1.98 | -0.57 | -0.72 – -0.43 |
| $100K to $150K | -1.22 | -1.86 – -0.57 | -2.72 | -3.39 – -2.05 | -3.58 | -4.58 – -2.58 | -0.7 | -0.84 – -0.56 |
| >$150K | -1.68 | -2.32 – -1.04 | -3.49 | -4.16 – -2.83 | -4.58 | -5.57 – -3.58 | -0.81 | -0.95 – -0.67 |
| **Gender** | | | | | | | | |
| Women | *Reference** | | *Reference** | | *Reference** | | *Reference** | |
| Men | -1.39 | -1.70 – -1.08 | -1.19 | -1.51 – -0.87 | -2.26 | -2.73 – -1.78 | -0.22 | -0.29 – -0.15 |
| Gender Diverse | 1.8 | 0.86 – 2.74 | 2.29 | 1.34 – 3.24 | 3.44 | 1.99 – 4.90 | 0.34 | 0.13 – 0.54 |
| **Increased alcohol use** | 0.08 | -0.20 – 0.36 | 0.07 | -0.21 – 0.36 | -0.38 | -0.83 – 0.08 | -0.03 | -0.10 – 0.03 |
| **Phase x Increased alcohol use** | | | | | | | | |
| Pre-COVID | *Reference** | | *Reference** | | *Reference** | | *Reference** | |
| Phase 1 | 1.12 | 0.89 – 1.35 | 1.06 | 0.83 – 1.29 | 2.36 | 1.94 – 2.78 | 0.16 | 0.10 – 0.22 |
| Phase 2/3 | 0.79 | 0.54 – 1.03 | 0.9 | 0.66 – 1.15 | 1.76 | 1.31 – 2.21 | 0.13 | 0.07 – 0.20 |
| Phase 2/3_2 | 0.79 | 0.31 – 1.27 | 0.77 | 0.29 – 1.26 | 1.67 | 0.80 – 2.55 | 0.14 | 0.02 – 0.27 |
| Phase 4 | 0.98 | 0.51 – 1.46 | 0.72 | 0.24 – 1.21 | 1.93 | 1.05 – 2.80 | 0.15 | 0.03 – 0.28 |
| Phase 5 | 1.23 | 0.75 – 1.70 | 1.07 | 0.58 – 1.55 | 2.18 | 1.30 – 3.05 | 0.08 | -0.04 – 0.20 |
| **Marginal R$^2$ /Conditional R$^2$** | 0.175 / 0.719 | | 0.165 / 0.722 | | 0.259 / 0.678 | | 0.148 / 0.624 | |

*CI* refers to confidence intervals for the adjusted estimates. Pre-COVID: Prior to mid-March 2020; Phase 1: Mid-March 2020 to mid-May 2020; Phase: 2/3: Mid-May 2020 to November 2020; Phase 2/3_2: Mid-May 2020 to August 2020; Phase 4: September 2020 to October 2020; Phase 5: November 2020 to March 1, 2021. GAD-7: Generalized Anxiety Disorder measure; PHQ-9: Patient Health Questionnaire; CRISIS: CoRonavIruS Health Impact Survey; Loneliness was measured using a single item.

*$p < .0001$.

## Associations between psychosocial outcomes and cannabis use by gender

A total of 5.9% of the sample reported an increase in cannabis use since the start of the pandemic. Increased cannabis use was negatively associated with age ($p < .001$, Table 3), with older participants having lower odds of increased use. There was a significant relationship with gender ($p = .02$, Table 3, Fig 6), where women had a significantly higher odds of increased cannabis use compared to men, and there was no significant difference between men and gender diverse, and women and gender diverse groups.

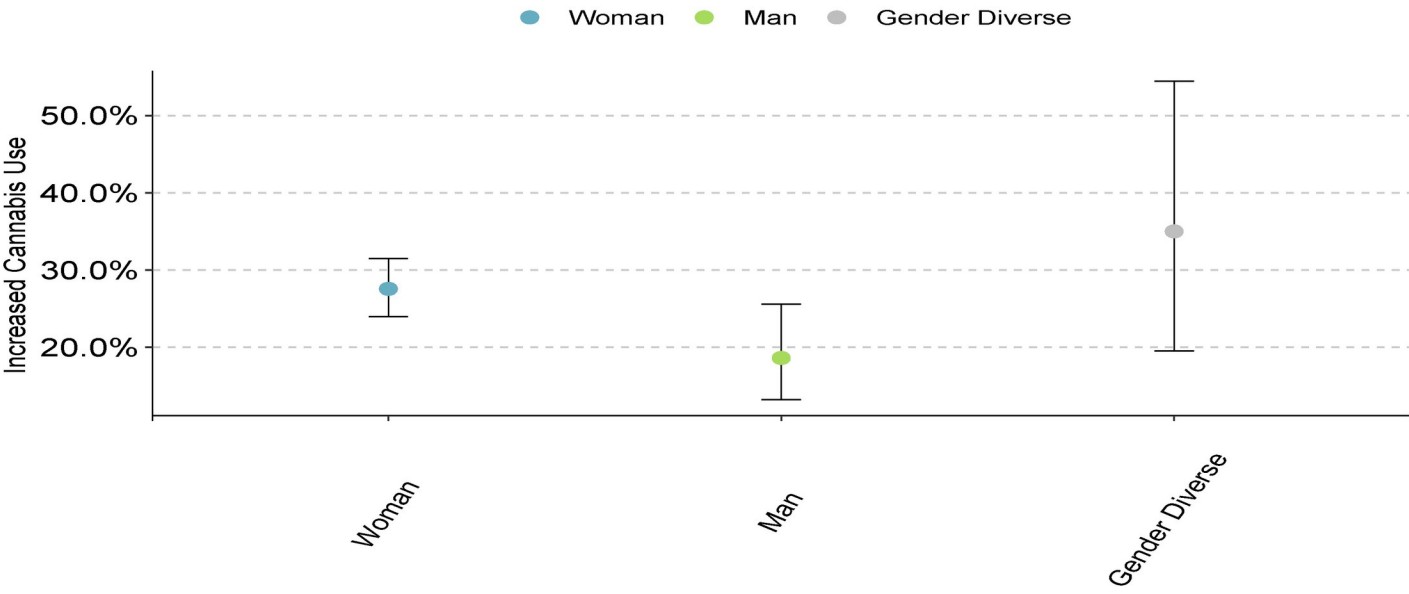

**Fig 6. Cannabis use across different genders.**

Controlling for household income, there was a significant interaction between change in cannabis use and pandemic phase ($p < .0001$ for GAD-7, PHQ-9, and CRISIS, $p = .04$ for Loneliness, Table 5). Post-hoc pairwise tests suggest that across all phases, including pre-COVID, those who increased cannabis use had significantly higher anxiety, more depressive symptoms, and higher COVID-stress scores than those who did not have increased cannabis use. Loneliness scores were significantly higher across all phases of the pandemic for those who increased cannabis use compared to those who did not. There was no interaction between gender and increased cannabis use for GAD-7, PHQ-9, or CRISIS scores. However, there was a significant interaction between gender and increased cannabis use on Loneliness ($p = .008$, Table 5). For both men and women, those who increased cannabis use had more loneliness symptoms than those who did not have increased cannabis use. Conversely, among the gender-diverse participants, there was no difference in loneliness between those who increased cannabis, and those who did not.

## Discussion

This large Canadian study recruited 6,076 women, men, and gender diverse people across the province of British Columbia. Our main findings indicated that age, sex, gender, ethnicity, Indigenous status, sexual orientation, and phase of the pandemic have distinct effects on psychosocial outcomes. Across outcomes, women had more symptoms of depression, anxiety, loneliness, and stress than men, regardless of their age or ethnicity, while the gender diverse group (n = 72) had even more symptoms than women. An analysis by sex revealed the same findings as for gender, except that the gender diverse group was now absorbed into one of the two binary sex categories and obscuring their findings.

Our results highlight the greater negative outcomes on all psychosocial variables in gender diverse individuals, which would have been obscured in an analysis by sex alone and adds to the literature highlighting the value in analyzing data by gender. It is important to underscore that being a woman was a significant factor that determined higher anxiety, depression, stress, and loneliness—a finding mirrored in the literature across all continents [38,39]. The novelty

**Table 5. Psychosocial outcomes and sociodemographic factors for those that reported an increase in cannabis.**

| Predictors | Anxiety (GAD-7) Estimates | Anxiety (GAD-7) 95% CI | Depression (PHQ-9) Estimates | Depression (PHQ-9) 95% CI | Pandemic Stress (CRISIS) Estimates | Pandemic Stress (CRISIS) 95% CI | Loneliness Estimates | Loneliness 95% CI |
|---|---|---|---|---|---|---|---|---|
| (Intercept) | 8.29*** | 7.56 – 9.03 | 9.02*** | 8.26 – 9.77 | 28.81*** | 27.67 – 29.95 | 2.51*** | 2.35 – 2.67 |
| Age (per year increase) | -0.09*** | -0.10 – -0.08 | -0.07*** | -0.08 – -0.06 | -0.14*** | -0.15 – -0.12 | -0.01*** | -0.01 – -0.01 |
| **Phase** | | | | | | | | |
| Pre-COVID | Reference*** | | Reference*** | | Reference*** | | Reference*** | |
| Phase 1 | 2.82 | 2.72 – 2.93 | 2.29 | 2.19 – 2.39 | 7.2 | 7.02 – 7.39 | 0.6 | 0.58 – 0.63 |
| Phase 2/3 | 2 | 1.89 – 2.11 | 1.9 | 1.79 – 2.01 | 5.23 | 5.03 – 5.43 | 0.46 | 0.44 – 0.49 |
| Phase 2/3_2 | 1.5 | 1.27 – 1.72 | 1.44 | 1.21 – 1.67 | 4.2 | 3.79 – 4.61 | 0.41 | 0.35 – 0.47 |
| Phase 4 | 2 | 1.78 – 2.23 | 2.26 | 2.03 – 2.49 | 5.85 | 5.44 – 6.26 | 0.62 | 0.56 – 0.67 |
| Phase 5 | 2.38 | 2.16 – 2.61 | 2.81 | 2.59 – 3.04 | 7.14 | 6.72 – 7.55 | 0.89 | 0.83 – 0.95 |
| **Household Income** | | | | | | | | 0. |
| <$10K to $20K | Reference*** | | Reference*** | | Reference*** | | Reference*** | |
| $20K to $40K | 0.44 | -0.30 – 1.17 | -0.68 | -1.44 – 0.08 | -0.43 | -1.57 – 0.72 | -0.21 | -0.37 – -0.05 |
| $40K to $60K | -0.04 | -0.72 – 0.65 | -1 | -1.71 – -0.29 | -1.23 | -2.31 – -0.16 | -0.31 | -0.46 – -0.16 |
| $60K to $80K | -0.65 | -1.34 – 0.03 | -1.95 | -2.65 – -1.24 | -2.61 | -3.67 – -1.55 | -0.48 | -0.63 – -0.33 |
| $80K to $100K | -1.02 | -1.68 – -0.35 | -2.42 | -3.11 – -1.74 | -2.98 | -4.02 – -1.95 | -0.57 | -0.71 – -0.42 |
| $100K to $150K | -1.11 | -1.75 – -0.47 | -2.62 | -3.28 – -1.96 | -3.45 | -4.44 – -2.45 | -0.69 | -0.83 – -0.55 |
| >$150K | -1.51 | -2.14 – -0.87 | -3.33 | -3.98 – -2.67 | -4.36 | -5.34 – -3.37 | -0.79 | -0.93 – -0.65 |
| **Gender** | | | | | | | | |
| Women | Reference*** | | Reference*** | | Reference*** | | Reference*** | |
| Men | -1.42 | -1.73 – -1.12 | -1.22 | -1.54 – -0.91 | -2.27 | -2.75 – -1.80 | -0.24 | -0.31 – -0.17 |
| Gender Diverse | 1.6 | 0.67 – 2.52 | 2.08 | 1.14 – 3.02 | 3.17 | 1.72 – 4.62 | 0.34 | 0.12 – 0.57 |
| **Increased cannabis use** | 1.2 | 0.71 – 1.70 | 1.29 | 0.78 – 1.80 | 1.69 | 0.88 – 2.50 | 0.06 | -0.06 – 0.18 |
| **Phase x Increased cannabis use** | | | | | | | | |
| Pre-COVID | Reference*** | | Reference*** | | Reference*** | | Reference* | |
| Phase 1 | 1.4 | 1.00 – 1.81 | 1.47 | 1.05 – 1.88 | 1.64 | 0.89 – 2.39 | 0.15 | 0.04 – 0.26 |
| Phase 2/3 | 1.43 | 0.97 – 1.89 | 1.53 | 1.07 – 2.00 | 1.85 | 1.01 – 2.69 | 0.14 | 0.02 – 0.26 |
| Phase 2/3_2 | 1.38 | 0.68 – 2.08 | 1.34 | 0.62 – 2.06 | 2.48 | 1.19 – 3.77 | 0.16 | -0.03 – 0.34 |
| Phase 4 | 1.8 | 1.11 – 2.50 | 2 | 1.28 – 2.71 | 2.62 | 1.34 – 3.91 | 0.18 | -0.01 – 0.36 |
| Phase 5 | 1.73 | 1.03 – 2.43 | 1.94 | 1.23 – 2.66 | 2.17 | 0.88 – 3.46 | 0.22 | 0.03 – 0.40 |
| **Gender x Increased cannabis use** | | | | | | | | |
| Women | | | | | | | Reference** | |
| Man | | | | | | | 0.45 | 0.16 – 0.73 |
| Gender Diverse | | | | | | | -0.14 | -0.70 – 0.42 |
| **Marginal R² /Conditional R²** | 0.183 / 0.718 | | 0.175 / 0.722 | | 0.262 / 0.675 | | 0.151 / 0.623 | |

*CI* refers to confidence intervals for the adjusted estimates. Pre-COVID: Prior to mid-March 2020; Phase 1: Mid-March 2020 to mid-May 2020; Phase: 2/3: mid-May 2020 to November 2020; Phase 2/3_2: mid-May 2020 to August 2020; Phase 4: September 2020 to October 2020; Phase 5: November 2020 to March 1, 2021. GAD-7: Generalized Anxiety Disorder measure; PHQ-9: Patient Health Questionnaire; CRISIS: CoRonavIruS Health Impact Survey; Loneliness was measured using a single item.

*p = .04

**p = .008

***p < .0001.

of this study, however, is that this effect of being a woman was not impacted by participants' age, ethnicity, or other sociodemographic variables. In other words, having a woman gender was sufficient to place individuals at higher risk for depression, anxiety, stress, and loneliness

over the pandemic. Given that women and gender diverse individuals are more likely to be diagnosed with mood disorders or score lower on mood surveys outside of a pandemic [16,40,41], it is not surprising that these populations are experiencing mental health inequities during COVID-19. However, our results should be interpreted with some caution as our gender-diverse cohort accounted for only 1% of the sample. Nonetheless, our results are striking and consistent with many other studies focused on gender using larger cohorts [42,43].

Our study also benefited from examining the effects of other sociodemographic variables, such as age, to determine how they might play a role in the effect of sex and gender on mental health. Across all the psychosocial measures, younger participants were more likely to have anxiety, depression, pandemic stress, and loneliness, irrespective of their gender. These findings are consistent with others in smaller cohort studies that indicated younger ages were associated with more psychosocial symptoms [44]. There may be several reasons for these findings such as restricted social engagements, barriers to employment, and living conditions. Lockdowns across the globe have resulted in restricted social gatherings, closing of restaurants, bars and clubs, as well as recreational sporting activities (gyms, sports clubs, exercise classes, yoga and dance). In addition, younger adults are more likely to either live on their own, or with unrelated roommates and have greater perceived lack of social support. Indeed, findings from a larger cohort in China found that greater loneliness was associated not only with younger age (16–29) but also in unmarried individuals [36]. Physical activity is another important factor as a large survey across fourteen countries found that decreased physical activity during restrictions and lockdowns, as well as high physical activity pre-pandemic, were associated with poorer mental health scores [45]. Other studies have also noted that suicide and suicidal ideation have increased during the pandemic in younger adults [46], related partially to job losses. Taken together, the underlying reasons for this significant effect of age are of great importance and require further study. At a minimum, these findings suggest that mental health resources tailored to younger individuals are required in any pandemic relief measures taken by government. It might not be sufficient to increase all mental health supports, but rather have tailored ones to young adults that are cost effective and accessible.

In addition to age, ethnicity was associated with psychosocial outcomes with Chinese/Taiwanese participants reporting significantly lower scores (i.e., fewer psychosocial symptoms) on anxiety, depression, pandemic stress, and loneliness. These data are consistent with findings from other studies, such as a survey of more than 46,000 Canadians which found that Chinese individuals were less likely to report symptoms consistent with moderate to severe generalized anxiety disorder than other visible minority groups during the COVID-19 pandemic [47]. It is possible that the lower rates of mental disorders seen in Asian or Chinese immigrants [48] may be due to cultural stigma associated with mental illness leading to lower rates of disclosure of psychological symptoms [48,49]. It is also possible that the lower rates of psychological symptoms may be due to differences in the validity of these measures cross culturally [34,50], leaving open the possibility of a measurement bias [51], although it was concluded to be a reliable measure across some cultural groups [51,52]. In sum, our findings suggest that our Chinese/Taiwanese sample experienced fewer psychosocial symptoms throughout the pandemic relative to other groups, and of note, ethnicity did not interact with gender or income to impact these outcomes. Similarly, gender did not interact with income to impact these outcomes.

We found that those who self-identified as Indigenous had significantly more psychosocial symptoms than non-Indigenous participants across all four scales for all phases of the pandemic in BC. Importantly, there was no difference in psychosocial outcomes between Indigenous and non-indigenous groups pre-COVID, which underscores the disproportionate impact of the pandemic on this community. While investigations on the mental health impacts

on Indigenous peoples during the COVID-19 pandemic have been limited, our results are consistent with the available data. For example, other data from Australia (Aboriginal or Torres Strait Islander) [53] as well as Canada [54] showed more psychosocial symptoms among Indigenous respondents during the COVID-19 pandemic. The lack of interaction between Indigenous status and gender suggests that the higher psychosocial symptoms occur regardless of an Indigenous persons' gender, standing in contrast to another study finding that Indigenous women were particularly impacted by mental health issues (severe generalized anxiety, worse mental health, and stress) during COVID-19 [54]. Future studies should explore the extent to which variables such as rurality (which can contribute to barriers accessing care) and income may account for these higher rates of psychological symptoms among Indigenous communities [44]. In the meantime, these findings point to the need for culturally-safe mental health resources being made available to Indigenous communities in any COVID relief efforts.

Findings on the relationship between anxiety, depression, pandemic stress, and loneliness, with increased alcohol and cannabis use, align with previous studies [12]. Given the poorer self-reported mental health among younger populations, it was not surprising to observe an increase in alcohol and cannabis use among this group, which suggests that alcohol use may be a form of coping for younger persons. We cannot attribute directionality to this association, nor eliminate the possibility that increased alcohol and cannabis use may be contributing to the increased psychosocial symptoms observed among younger populations during the pandemic. The lack of a gender difference in increased alcohol use is in contrast with a previous American study [55] which found that females had increased their alcohol use compared to males. It may be that differences in the samples accounts for these contrasting findings. It is also possible that the increase seen in men in our sample was higher than in previous studies, thus rendering the gender difference void. In contrast, we saw a gender effect on increased cannabis use, which was expressed by women, but not by men or gender diverse persons. In recent surveys, 28% of British Columbians had engaged in cannabis use in the past twelve months, compared to the Canadian average of 11%, suggesting that British Columbians are more likely to engage in cannabis use, and therefore may be more likely to use cannabis as a form of coping [56,57]. Although cannabis use has been associated with male typicality and may go against gender norms typical to women [58], it may be that the social isolation disrupted these social norms and facilitated women's more active engagement in additional cannabis use, relative to pre-pandemic levels.

Our findings align with the global trend of increased substance use, as recent studies have demonstrated that alcohol, cannabis, and opiate use changed during and post-lockdown [59]. Alcohol use has remained elevated relative to pre-pandemic levels, and though opiate use seemed to have dropped during lockdowns, a return to regular dosage post-lockdown has helped to drive overdoses, due to diminished tolerance [59]. It is possible that deteriorated mental health could be attributed to overuse of certain substances, though studies with multiple follow-up points are needed to determine a causal pathway for the increase in psychosocial symptoms demonstrated in our study. Future studies should aim to elucidate potential mechanisms by which substance use can influence mental health in the context of a pandemic and lockdowns to mitigate the consequences of public health interventions on well-being.

As predicted, psychosocial symptoms worsened over the course of the pandemic, with some of the highest symptoms observed early on, aligning with previous studies that found a higher prevalence of mental health disorders during the initial COVID-19 lockdown in March 2020 [60,61]. Phases 2 and 3 of COVID restrictions in BC were characterized by an easing of restrictions, permitting outdoor gatherings and small social events, and the summer season. This loosening of public health measures was associated with a slight improvement in mental health, more than likely due to an increase in perceived social support and optimism regarding

the state of the pandemic. Mental health outcomes then worsened in Phases 4 and 5 as BC entered wave 2 of the pandemic and public health orders tightened once again. It is important to note the average PHQ-9 and GAD-7 scores did not meet the criteria for clinical depression or anxiety, but that these levels increased relative to pre-pandemic levels as well as over time.

## Strengths and limitations

Our study benefitted from a large, population-based sample size, and, despite known mental health disparities by gender, as far as we are aware, was one of the few that sought to explore findings from a gender lens by including gender-diverse groups as well, given known mental health disparities by gender [17]. That said, our sample size for gender diverse individuals was still limited [17]. Future studies should further investigate mental health in the gender diverse community during the COVID-19 pandemic with a focus on people of all ages, in contrast to previous studies [22]. Another limitation of the present study was the retrospective, cross-sectional nature of the survey, where participants completed the survey at only one time point, and were asked to retrospectively recall their mood and anxiety during different time points. This may have increased the likelihood of recall bias and reducing our capacity to examine causality and directionality of poor mental health outcomes. Finally, this study was confined to the general population of BC, and only individuals who had access to email and internet, and therefore results may only be generalizable to the Canadian population, and populations with similar demographics to the present study, and to individuals who have access to email and internet.

## Implications

Our study has important implications for public health policy. These findings illustrate that government policies and interventions for future pandemics should place on emphasis on young adults, low-income populations, women, Indigenous, and gender diverse communities. Additionally, our study was one of the first to measure mental health outcomes across different phases of the pandemic, directly examining the effect of increased public health measures on mental health. At the time of writing, the vaccine rollout is well underway in BC with experts predicting an end to the pandemic in the months ahead, however, it is unclear whether mental health will return to pre-pandemic levels, or when life will return to "normal." Moving forward, policy makers and leaders need to consider our findings when planning future public health measures. In future pandemics, the mental health of marginalized populations needs to be considered proactively. As vaccination efforts continue and case counts fall, it will also be critical to monitor the health status of these populations to ensure that they are not left behind. Additionally, for future pandemics and outbreaks, mobilizing resources to these communities early on can aid in mitigating these inequities from the beginning, rather than as an afterthought.

## Supporting information

**S1 Table. Public health measures during different phases of the COVID-19 pandemic in British Columbia.** Measures listed are not exhaustive.
(DOCX)

## Acknowledgments

We wish to thank Falla Jin and Shanlea Gordon (both at the BC Children's Research Institute) for their assistance with data collection.

## Author Contributions

**Conceptualization:** Lori A. Brotto, Angela Kaida, Gina S. Ogilvie, Liisa Galea.

**Data curation:** Arianne Albert.

**Formal analysis:** Arianne Albert.

**Funding acquisition:** Lori A. Brotto, Gina S. Ogilvie, Liisa Galea.

**Investigation:** Lori A. Brotto, Arianne Albert, Angela Kaida, Manish Sadarangani, Gina S. Ogilvie, Liisa Galea.

**Methodology:** Lori A. Brotto, Alexandra Baaske, Arianne Albert, Amy Booth, Angela Kaida, Laurie W. Smith, Sarai Racey, Anna Gottschlich, Manish Sadarangani, Gina S. Ogilvie, Liisa Galea.

**Project administration:** Lori A. Brotto, Alexandra Baaske, Gina S. Ogilvie.

**Supervision:** Lori A. Brotto, Alexandra Baaske, Angela Kaida, Gina S. Ogilvie, Liisa Galea.

**Writing – original draft:** Lori A. Brotto, Kyle Chankasingh, Alexandra Baaske, Arianne Albert, Angela Kaida, Gina S. Ogilvie, Liisa Galea.

**Writing – review & editing:** Lori A. Brotto, Kyle Chankasingh, Alexandra Baaske, Arianne Albert, Amy Booth, Angela Kaida, Laurie W. Smith, Sarai Racey, Anna Gottschlich, Melanie C. M. Murray, Manish Sadarangani, Gina S. Ogilvie, Liisa Galea.

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
