## [Decision Letter · Decision Letter 0]

21 Sep 2021

PONE-D-21-18028The influence of sex, gender, age, and ethnicity on psychosocial factors and substance use throughout phases of the COVID-19 pandemicPLOS ONE

Dear Dr. Brotto,

Thank you for submitting your manuscript to PLOS ONE. After careful consideration, we feel that it has merit but does not fully meet PLOS ONE’s publication criteria as it currently stands. Therefore, we invite you to submit a revised version of the manuscript that addresses the points raised during the review process.

Two reviewers provided feedback on this interesting paper.  Reviewer 1 identified numerous methodological issues and recommended further focus and less breadth in the discussion.  In contrast Reviewer 2 indicated that the scope could be widened.  In appreciation of both reviews, I gave this a careful read.  I agree with Reviewer 1 that the paper should be revised substantially in all areas identified by that reviewer.  Please ensure that your decision is justified on PLOS ONE’s publication criteria and not, for example, on novelty or perceived impact.

We look forward to receiving your revised manuscript.

Kind regards,

Kimberly Page, PhD, MPH

Academic Editor

PLOS ONE

Journal Requirements:

2. Peer review at PLOS ONE is not double-blinded (https://journals.plos.org/plosone/s/editorial-and-peer-review-process). For this reason, authors should include in the revised manuscript all the information removed for blind review.

Reviewers' comments:

Reviewer's Responses to Questions

**Comments to the Author**

1. Is the manuscript technically sound, and do the data support the conclusions?

Reviewer #1: Yes

Reviewer #2: Yes

2. Has the statistical analysis been performed appropriately and rigorously? 

Reviewer #1: Yes

Reviewer #2: Yes

3. Have the authors made all data underlying the findings in their manuscript fully available?

Reviewer #1: Yes

Reviewer #2: Yes

4. Is the manuscript presented in an intelligible fashion and written in standard English?

Reviewer #1: Yes

Reviewer #2: Yes

5. Review Comments to the Author

Reviewer #1: “The influence of sex, gender, age, and ethnicity on psychosocial factors and substance use throughout phases of the COVID-19 pandemic” applies gender and intersection lenses to an investigation of mental health and substance use. It includes over 6,000 people living in Canada, the majority of whom are white (81%), heterosexual (85%), women (87%), living in an urban area (94%), who earn over $80,000 per year (57%) and have access to email as well as the internet (100% by virtue of recruitment). Authors report that outcomes are significantly associated with pandemic phase, gender, and age.

This is an important topic and little research has been conducted in regard to the COVID-19 pandemic. The authors report interesting findings, which could help inform responses to future pandemics. Three points dampen my enthusiasm for the paper. First, sampling methods appear to be haphazard, making it difficult to understand the results and to whom they can be generalized. Second, the large number of outcomes renders the discussion as rather surface and lacking an anchor or main “thread.” Third, the novel findings discovered here aren’t highlighted in a way that makes them stand out; how they advance the field, and how they should be used need to be further developed.

A few suggestions:

-The title and abstract refer to a gender lens and the introduction refers to an intersectional lens. The introduction then goes on to motivate a gender-based study, but brings back intersectionality in paragraph 5. It would be helpful for the authors to commit and focus the paper on either gender or intersectionality (which includes gender, but is a different topic and needs to be set up differently). If intersectionality is be the focus of the paper, it should be defined and the paper will need some editing. Intersectionality is generally considered the interaction between aspects of social status (gender, race, socioeconomic status), which requires a different introduction (different motivation). For example, ethnicity is a major player in intersectionality, but the authors include only one sentence on minority stress in the introduction (and as an “other social determinant of health), with nothing about how race, sex and socioeconomic status have been shown in prior research to influence outcomes studied here. Also, if the paper is focused on intersectionality, interaction effects should be the main effects of interest.

-The authors introduce a lot of topics in the introduction which makes the the point of the paper unclear. It would be helpful for the authors to streamline the text a bit. Also, while a variety of topics are raised, several key topics are left out. For example, each outcome is a huge topic in itself and to understand how this study advances science, the reader needs to understand what is already known about the outcomes (e.g., why are the questions asked here important for each outcome?) Understanding what results reported here mean will require at least some background on each outcome (or background on considering them collectively). This may be a lot to accomplish for so many outcome variables.

-“Phases” of the pandemic are mentioned, but what they are and why they matter is not specified. Are the dates the authors assigned to phases arbitrary? Exact what does each phase represent? The authors hypothesize that “during phases of increased social restriction psychosocial symptoms would increase,” suggesting that social restriction somehow contributes to the definition of each phase, but exactly how that happens is unclear.

-More detail on inclusion criteria, sampling methods and recruitment venues of the original studies is needed.

-The statistical analysis section states that longitudinal analyses were conducted, but a limitation of the study cited by authors is “cross-sectional nature of the survey, where participants completed the survey at only one time point.” Please clarify.

-There are multiple types of longitudinal data analysis so please be more specific about analytic methods.

-The abstract lists six outcomes, but the last paragraph of the introduction seems to suggest only four mental health outcomes, with analyses that consider how these outcomes are associated with the substance use variables. This may just be a matter of a few edits to the final sentences of the introduction, but please be specific and consistent throughout the manuscript to improve clarity.

-The last paragraph of the introduction also refers to the four “psychosocial domains.” Please explain how the four mental health variables map onto psychosocial domains. Referring to a score intended to identify a mental disorder as a domain is confusing, as is some of the text around it (e.g., “We predicted that these four psychosocial domains would be more symptomatic in women and gender diverse individuals”).

-Given that potential participants were invited to enroll in the current study via email and to take an online survey, people without email or access to the internet were left out. This overlooked low-income individuals, which is a limitation of the study and should be cited as such.

-Authors state “To improve the representativeness of the study sample, Index Participants were asked to pass the invitation on to one household member who identified as a different gender as the respondent.” While this would have increased the sample size, and possibly increase gender variability, it would not necessarily increase representativeness of the population. To the contrary, sampling from the same household results in another person who is very similar to the first in terms of race and socioeconomic factors. In the results section, it becomes clear that this population is in fact composed of many more women (87%) than men. Why the population is so skewed toward women deserves explanation.

-The authors do not appear to have used a measure of loneliness that was previously tested for reliability and validity. It would be helpful for the authors to provide assessment of reliability and validity for the measure they created here.

-A major problem with this study is understanding exactly who it represents. It starts out including people who have participated in prior research (but recruitment criteria are not provided). It then goes on to state that, after two months of data collection, recruitment was expanded. Why expansion occurred is unclear and how the additional sampling venues were chosen is similarly unclear. Public recruitment through social media, and “engagement of community groups and other stakeholders” (without explanation as to what is meant by stakeholder in this specific scenario) make it very difficult to understand who this study applies to and what it means. The number recruited using each sampling mode is not stated in the results, and there is also no attempt to account for recruitment mode in analyses.

-On average, the sample seems to be college-educated, straight, white women who live in an urban environment, have access to the internet, and earn over $80,000 a year. This isn’t entirely clear and it would be helpful to distill information like this in a summarized fashion to give the reader an idea of who is included.

-Regarding lines 152 and 153: how was the target sample size of 750 determined?

-Discussing mental health scores is confusing (e.g., lower usually indicates worse health, but in this case, it refers to fewer symptoms and better health). Consider referring to the number of symptoms throughout instead of scores.

-The text refers to symptom scores (suggesting a continuous variable), but the table simply states each mental health condition (suggesting a dichotomous variable). Given that the methods section describes both formats, this is confusing. Consider clarifying the format of the table variables on the table.

-The text refers to sex and gender separately and I’m assuming that the headers in Table 1 also refer to gender, but it would be helpful to specify that “women” and “men” refers to gender and not sex. Given the focus on gender and sex in this paper, it would also be helpful to add a row showing how sex breaks down by gender.

-The methods section states, “We included pairwise interactions to assess non additive effects between pandemic phase, sex, gender, ethnicity, sexual orientation, income, and Indigenous status.” However, the results section leads with a series of interactions based on age. There is a disconnect here. Also, intersectionality is often thought of as being based in race and gender, so the results section was surprising due to its inconsistency with both earlier methods text (which did not emphasize age) and the field of study (which often focuses on gender and race). Age is certainly an important factor; so authors might want to consider (1) developing the introduction in a way that talks about age as a component of intersectionality, and (2) being consistent between methods and results in terms of factors discussed.

-Unless I missed it, there were no significant interactions between race and sex or race and income or sex and income with regard to the mental health and substance use outcomes. It may be helpful to clearly state this in the discussion.

-The discussion is rather surface and fails to highlight what is new here and how, in combination with prior research, results could be used to improve policy or practice (i.e., beyond the fact that it should be used, exactly how should it be used?) For example

●The discussion states that women were more likely to have worse mental health scores and this is consistent with prior studies. How was this study different, novel or unique; how do the novel findings contribute something new here; and how should policy or practice be altered based on this study?

●There is a paragraph devoted to why age may have been associated with study outcomes, but I wonder if the authors have any suggestions for changes to pandemic-related policy or practice based on age? Or how future research could use this finding?

● The discussion states that this study, which found no gender effect on alcohol use, is in contrast to a prior study. Why might that be? Geographic differences? Differences in the population? Differences in terms of the time point (relative to the beginning of the pandemic) studied?

● The authors state, “These findings illustrate that government policies and interventions for future pandemics should place on emphasis on young adults, low-income populations, women, Indigenous, and gender diverse communities.” I wonder what this means and how it would be done. It would be extremely helpful for the authors to cite one or two pandemic policies and suggest exactly how they could have placed an emphasis on the factors identified here.

● Similar to the prior bullet point, the final conclusion of the abstract is, “Our findings highlight the need for policy makers and leaders to proactively consider gender when tailoring public health measures for future pandemics,” which is rather generic. It could have been written (based on prior COVID research) before this study was conducted. What do these results uniquely suggest that policy makers should do specifically?

-The paragraph regarding stigma against Chinese culture and validity of study measures in Chines Canadians is long and seems to be a tangent. Consider a more succinct paragraph on how these results harmonize with prior studies and how findings could be put to practice.

-Authors state that there was a trend for increasing odds of increased alcohol use as household income increased. However, given that none of the income categories were significantly different than the reference, this seems unlikely and the statement needs justification (e.g., was the trend statistically significant?)

-The abstract states that participants were asked questions “across five phases of the pandemic as well as retrospectively before the pandemic,” which seems to suggest multiple measures. The text should be edited to clearly indicate that this is a cross-sectional study.

-Consider “fewer symptoms” rather than “less symptoms.”

-Consider additional editing (e.g., using “have” multiple times in the same sentence; also, regarding Lines 84 and 85, rather than stating that age may interact with sex and psychosocial outcomes [suggesting a 3-way interaction], I think the authors mean that age may interact with sex to influence psychosocial outcomes).

Reviewer #2: Dear Authors,

I have perused with great interest your praiseworthy contribution titled The influence of sex, gender, age, and ethnicity on psychosocial factors and substance use throughout phases of the COVID-19 pandemic, which aims to elaborate on one of the most consequential aspects of the ongoing pandemic.

The article is well-written and competently assembled. The discussion is broad-ranging and thorough, its conclusions well supported by solid methodology.

Analyzing the influence of sex, gender, age and ethnicity confers an element of novelty to the report which makes it a valuable piece of research of considerable interest for the public at large. Discrepancies in how deeply the pandemic has impacted different segments of the population are well expounded upon, and that itself has value in terms of contributing to more effective policy-making approaches and strategies.

An element of weakness attributable to studies such as this has to do with its being cross-sectional, which makes it all but impossible to establish the actual correlations and causality directions taking into account all the study's variables. In that regard, future longitudinal studies will go a long way towards establishing actual causality. All data were self-reported, hence liable to be affected by well-established method biases, as you hinted.

Since the title mentions "substance use", I believe it would be advisable for the authors to dig a buit deeper than alcohol and cannabis, although their research has only accounted for those.

I would recommend making a brief subchapter or mention in the conclusions as to how substance use worldwide has been profoundly reshaped as a result of the pandemic, which might dovetail with some of the conclusions you arrived at based on your data, both from a substance use prspective and a psychosocial one. That would go a long way toward broadening the scope of your article, allowing for more in-depth comparisons of your own findings versus other sources.

You may want to draw upon the following sources:

Zaami S, Marinelli E, Varì MR. New Trends of Substance Abuse During COVID-19 Pandemic: An International Perspective. Front Psychiatry. 2020 Jul 16;11:700. doi: 10.3389/fpsyt.2020.00700.

Ornell F, Moura HF, Scherer JN, Pechansky F, Kessler FHP, von Diemen L. The COVID-19 pandemic and its impact on substance use: Implications for prevention and treatment. Psychiatry Res. 2020 Jul;289:113096. doi: 10.1016/j.psychres.2020.113096.

Stack E, Leichtling G, Larsen JE, Gray M, Pope J, Leahy JM, Gelberg L, Seaman A, Korthuis PT. The Impacts of COVID-19 on Mental Health, Substance Use, and Overdose Concerns of People Who Use Drugs in Rural Communities. J Addict Med. 2020 Nov 3:10.1097/ADM.0000000000000770. doi: 10.1097/ADM.0000000000000770.

Zaami S. New psychoactive substances: concerted efforts and common legislative answers for stemming a growing health hazard. Eur Rev Med Pharmacol Sci. 2019 Nov;23(22):9681-9690.

Lastly, since your manuscript qualifies as a research article, it would be best to have a structured abstract withObjectives, Methods and study design, Results and Conclusions.

All in all, the manuscript will make for a fine and valuable conytribution of great interest to a large audience.

Congratulations on your commendable effort.

6. PLOS authors have the option to publish the peer review history of their article (what does this mean?). If published, this will include your full peer review and any attached files.

Reviewer #1: No

Reviewer #2: No

---

## [Author Response · Author response to Decision Letter 0]

9 Oct 2021

9 October 2021

Editorial Team

PLOS One

Dear members of the editorial team, 

Thank you very much for your email dated September 21, 2021 that our paper, “The influence of sex, gender, age, and ethnicity on psychosocial factors and substance use throughout phases of the COVID-19 pandemic” was provisionally accepted pending a revised manuscript was reviewed and deemed adequate. 

We have carefully considered each of the two reviewers comments and addressed each out, as outlined point by point in the attached response letter. We have also included both a clean copy and a track changed copy.

Thank you very much for your ongoing consideration of our manuscript. 

Respectfully submitted,

Sincerely,

Lori A Brotto PHD, R PSYCH

Executive Director, Women’s Health Research Institute

Professor | Department of Obstetrics & Gynaecology, University of British Columbia

Canada Research Chair | Women’s Sexual Health 

Allied Staff Member | Vancouver Acute Health Service

---

## [Editor Report · Decision Letter 1]

25 Oct 2021

The influence of sex, gender, age, and ethnicity on psychosocial factors and substance use throughout phases of the COVID-19 pandemic

PONE-D-21-18028R1

Dear Dr. Brotto,

We’re pleased to inform you that your manuscript has been judged scientifically suitable for publication and will be formally accepted for publication once it meets all outstanding technical requirements.

Kind regards,

Kimberly Page, PhD, MPH

Academic Editor

PLOS ONE

Additional Editor Comments (optional):

I commend the authors on their responses to reviewers critiques and the very well revised manuscript. I do think this articles contributes important data and knowledge regarding gender differences and response to the COVID-19 pandemic.
---

## [Editor Report · Acceptance letter]

11 Nov 2021

PONE-D-21-18028R1 

The influence of sex, gender, age, and ethnicity on psychosocial factors and substance use throughout phases of the COVID-19 pandemic 

Dear Dr. Brotto:

I'm pleased to inform you that your manuscript has been deemed suitable for publication in PLOS ONE. Congratulations! Your manuscript is now with our production department. 

Kind regards, 

on behalf of

Dr. Kimberly Page 

Academic Editor

PLOS ONE